# CNApp, a tool for the quantification of copy number alterations and integrative analysis revealing clinical implications

Sebastià Franch-Expósito[1†], Laia Bassaganyas[2†], Maria Vila-Casadesús[3], Eva Hernández-Illán[1], Roger Esteban-Fabró[2], Marcos Díaz-Gay[1], Juan José Lozano[3], Antoni Castells[1], Josep Maria Llovet[2,4,5], Sergi Castellví-Bel[1]*, Jordi Camps[1,6]*

[1]Gastrointestinal and Pancreatic Oncology Team, Institut D'Investigacions Biomèdiques August Pi i Sunyer (IDIBAPS), Hospital Clínic de Barcelona, Centro de Investigación Biomédica en Red de Enfermedades Hepáticas y Digestivas (CIBEREHD), Universitat de Barcelona, Barcelona, Spain; [2]Liver Cancer Translational Research Group, Liver Unit, Institut D'Investigacions Biomèdiques August Pi i Sunyer (IDIBAPS), Hospital Clínic de Barcelona, Centro de Investigación Biomédica en Red de Enfermedades Hepáticas y Digestivas (CIBEREHD), Universitat de Barcelona, Barcelona, Spain; [3]Bioinformatics Unit, CIBEREHD, Barcelona, Spain; [4]Mount Sinai Liver Cancer Program, Division of Liver Diseases, Tisch Cancer Institute, Icahn School of Medicine at Mount Sinai, New York, United States; [5]Institució Catalana de Recerca i Estudis Avançats (ICREA), Barcelona, Spain; [6]Unitat de Biologia Cel·lular i Genètica Mèdica, Departament de Biologia Cel·lular, Fisiologia i Immunologia, Facultat de Medicina, Universitat Autònoma de Barcelona, Bellaterra, Spain

*For correspondence:
SBEL@clinic.cat (SC-B);
JCAMPS@clinic.cat (JC)

†These authors contributed equally to this work

**Abstract** Somatic copy number alterations (CNAs) are a hallmark of cancer, but their role in tumorigenesis and clinical relevance remain largely unclear. Here, we developed CNApp, a web-based tool that allows a comprehensive exploration of CNAs by using purity-corrected segmented data from multiple genomic platforms. CNApp generates genome-wide profiles, computes CNA scores for broad, focal and global CNA burdens, and uses machine learning-based predictions to classify samples. We applied CNApp to the TCGA pan-cancer dataset of 10,635 genomes showing that CNAs classify cancer types according to their tissue-of-origin, and that each cancer type shows specific ranges of broad and focal CNA scores. Moreover, CNApp reproduces recurrent CNAs in hepatocellular carcinoma and predicts colon cancer molecular subtypes and microsatellite instability based on broad CNA scores and discrete genomic imbalances. In summary, CNApp facilitates CNA-driven research by providing a unique framework to identify relevant clinical implications. CNApp is hosted at https://tools.idibaps.org/CNApp/.

## Introduction

The presence of somatic copy number alterations (CNAs) is a ubiquitous feature in cancer. In fact, the distribution of CNAs is sufficiently tissue-specific to distinguish tumor entities (*Ried et al., 2012*), and allows the identification of tumors responsive to particular therapies (*Cairncross et al., 2013*; *Davoli et al., 2017*). Moreover, high levels of CNAs, which result from chromosome instability, are generally associated with high-grade tumors and poor prognosis (*Hieronymus et al., 2018*; *Sansregret et al., 2018*; *Smith and Sheltzer, 2018*; *Stopsack et al., 2019*).

**eLife digest** In most cases, human cells contain two copies of each of their genes, yet sometimes this can change, an effect called copy number alteration (CNA). Cancer is a genetic disease and thus, studying the DNA from tumor samples is crucial to improving diagnosis and choosing the right treatment. Most tumors contain cells with CNAs; however, the impact of CNAs in cancer progression is poorly understood. CNAs can be studied by examining the genome of tumor cells and finding which regions display an unusual number of copies. It may also be possible to gather information about different cancer types by analyzing the CNAs in a tumor, but this approach requires the analysis of large amounts of data.

To aid the analysis of CNAs in cancer cells, Franch-Expósito, Bassaganyas et al. have created an online tool called CNApp, which is able to identify and count CNAs in genomic data and link them to features associated with different cancers. The hope is that a better understanding of the effect of CNAs in cancer could help better diagnose cancers, and improve outcomes for patients. Potentially, this could also predict what type of treatment would work better for a specific tumor. Besides, by using a machine-learning approach, the tool can also make predictions about specific cancer subtypes in order to facilitate clinical decisions.

Franch-Expósito, Bassaganyas et al. tested CNApp using previously existing cancer data from 33 different cancer types to show how CNApp can help the interpretation of CNAs in cancer. Moreover, CNApp can also use CNAs to identify different types of bowel (colorectal) cancer in a way that could help doctors to make decisions about treatment. Together these findings show that CNApp provides an adaptable and accessible research tool for the study of cancer genomics, which could provide opportunities to inform medical procedures.

Two main subtypes of CNAs can be discerned: broad CNAs, which are defined as whole-chromosome and chromosomal arm-level alterations, and focal CNAs, which are alterations of limited size ranging from part of a chromosome-arm to few kilobases (*Krijgsman et al., 2014*; *Zack et al., 2013*). Recently, it has been uncovered that while focal events mainly correlate with cell cycle and proliferation markers, broad aberrations are mainly associated with immune evasion markers, suggesting that tumor immune features might be determined by mechanisms related to overall gene dosage imbalance rather than specific actionable genes (*Buccitelli et al., 2017*; *Davoli et al., 2017*; *Taylor et al., 2018*). Furthermore, it has been shown that broad CNAs involving whole-chromosome arms may confer high risk of lethal disease in prostate cancer (*Stopsack et al., 2019*). Nevertheless, the precise role of CNAs in tumor initiation and progression, as well as their clinical relevance and therapeutic implications in most cancer types remain still poorly understood.

Interpretation and visualization of CNAs is time-consuming and very often requires complex analyses with clinical and molecular information. Well-established CNA algorithms, such as the gold-standard circular binary segmentation, define the genomic boundaries of copy number gains and losses based on signal intensities or read depth obtained from array comparative genomic hybridization and SNP-array or next-generation sequencing data, respectively (*Olshen et al., 2004*). However, the tumor-derived genomic complexity may cause an under- or overestimation of CNAs. This complexity is represented by tumor purity, tumor aneuploidy, and intratumor heterogeneity, which imply high levels of subclonal alterations. Thus, recent segmentation methods improved the accuracy to identify copy number segments in tumor samples either by considering the B allele frequency (BAF), such as ExomeCNV (*Sathirapongsasuti et al., 2011*), Control-FREEC (*Boeva et al., 2012*) and SAAS-CNV (*Zhang and Hao, 2015*), or through adjusting by sample purity and ploidy estimates, such as GAP (*Popova et al., 2009*), ASCAT (*Van Loo et al., 2010*) and ABSOLUTE (*Carter et al., 2012*). However, the state-of-the-art computational approach for CNA analysis in cancer is GISTIC2.0 (*Mermel et al., 2011*), which is a gene-centered probabilistic method that enables to define the boundaries of recurrent putative driver CNAs in large cohorts (*Beroukhim et al., 2010*). Nevertheless, despite ongoing progress on identifying CNAs, to our knowledge none of the existing software packages is readily available for integrative analyses to unveil their biological and clinical implications.

To address this issue, we developed CNApp, the first open-source application to quantify CNAs and integrate genomic profiles with molecular and clinical variables. CNApp is a web-based tool that provides the user with high-quality interactive plots and statistical correlations between CNAs and annotated variables in a fast and easy-to-explore interface. In particular, CNApp uses purity-corrected genomic segmented data from multiple genomic platforms to redefine CNA profiles, to compute CNA scores based on the number, length and amplitude of broad and focal genomic alterations, to assess differentially altered genomic regions, and to perform machine learning-based predictions to classify tumor samples. To exemplify the applicability and performance of CNApp, we used publicly available segmented data from The Cancer Genome Atlas (TCGA) to (i) measure the burden of global, broad, and focal CNAs as well as generate CNA profiles in a pan-cancer dataset spanning 33 cancer types, (ii) identify cohort-based recurrent CNAs in hepatocellular carcinoma and compare them with previously reported data, and (iii) assess predicting models for colon cancer molecular subtypes and microsatellite instability status based on CNA scores and specific genomic imbalances. CNApp is hosted at https://tools.idibaps.org/CNApp/ and the source code is freely available at GitHub (*Franch-Expósito, 2020* ; copy archived at https://github.com/elifesciences-publications/CNApp).

## Results

### Implementation

CNApp comprises three main sections: 1- *Re-Seg and Score: re-segmentation, CNA scores computation, variable association* and *survival analysis*, 2- *Region profile: genome-wide CNA profiling, CNA frequencies, correlation profiles* and *descriptive regions*, and 3- *Classifier model: machine learning classification model predictions* (*Figure 1*). Each of these sections and their key functions are described below. The input file consists of a data frame with copy number segments provided by any segmentation algorithm. Mandatory fields and column headers are sample name (*ID*), chromosome (*chr*), start (*loc.start*) and end (*loc.end*) genomic positions, and the log2 ratio of the copy number amplitude (*seg.mean*) for each segment. Section one incorporates the correction for tumor purity (i.e. fraction of tumor cells in the sample) to measure the actual magnitude of CNAs. Thus, when available, the input file will also include sample purity estimations (*purity*) and BAF values (*BAF*), which correct the accuracy of CNA calls and provide copy number neutral loss-of-heterozygosity (CN-LOH) events. Ploidy values, if known, might also be indicated as an independent variable. Annotation of variables can be included in the input file (tagged in every segment from each sample) or by uploading an additional file indicating new variables *per* sample.

### Re-Seg and score: re-segmentation, CNA scores computation, variable association and survival analysis

First, CNApp applies a re-segmentation approach to adjust for amplitude divergence due to technical variability and correct for estimated tumor purity. Default re-segmentation settings include *minimum segment length* (100 Kbp), *minimum amplitude (seg.mean) deviation from segment to zero* (0.16), *maximum distance between segments* (1 Mb), *maximum amplitude (seg.mean) deviation between segments* (0.16), and *maximum BAF deviation between segments* (0.1). These parameters can be customized by the user to better adjust the re-segmentation for each particular dataset. Re-segmented data are then used to calculate the broad (BCS), focal (FCS) and global (GCS) CNA scores, which provide three different quantifications of CNA levels for each sample. To compute these scores, CNApp classifies and weights CNAs based on their length and amplitude. For each sample, BCS is computed by considering broad (chromosome and arm-level) segment weights according to the amplitude value. Likewise, calculation of FCS takes into account weighted focal CNAs corrected by the amplitude and length of the segment. Finally, GCS is computed by considering the sum of normalized BCS and FCS, providing an overall assessment of the CNA burden.

To assess the reliability of CNA scores, we compared each score with the corresponding fraction of altered genome using a TCGA pan-cancer set of 10,635 samples. Both BCS (ranging from 0 to 44) and FCS (values ranging from 5 to 2466) highly correlated with the fraction of altered genome by broad and focal copy number changes, respectively (Spearman's rank correlation for BCS = 0.957 and for FCS = 0.938) (*Figure 1—figure supplment 1A and B—source data 1*). As expected, GCS

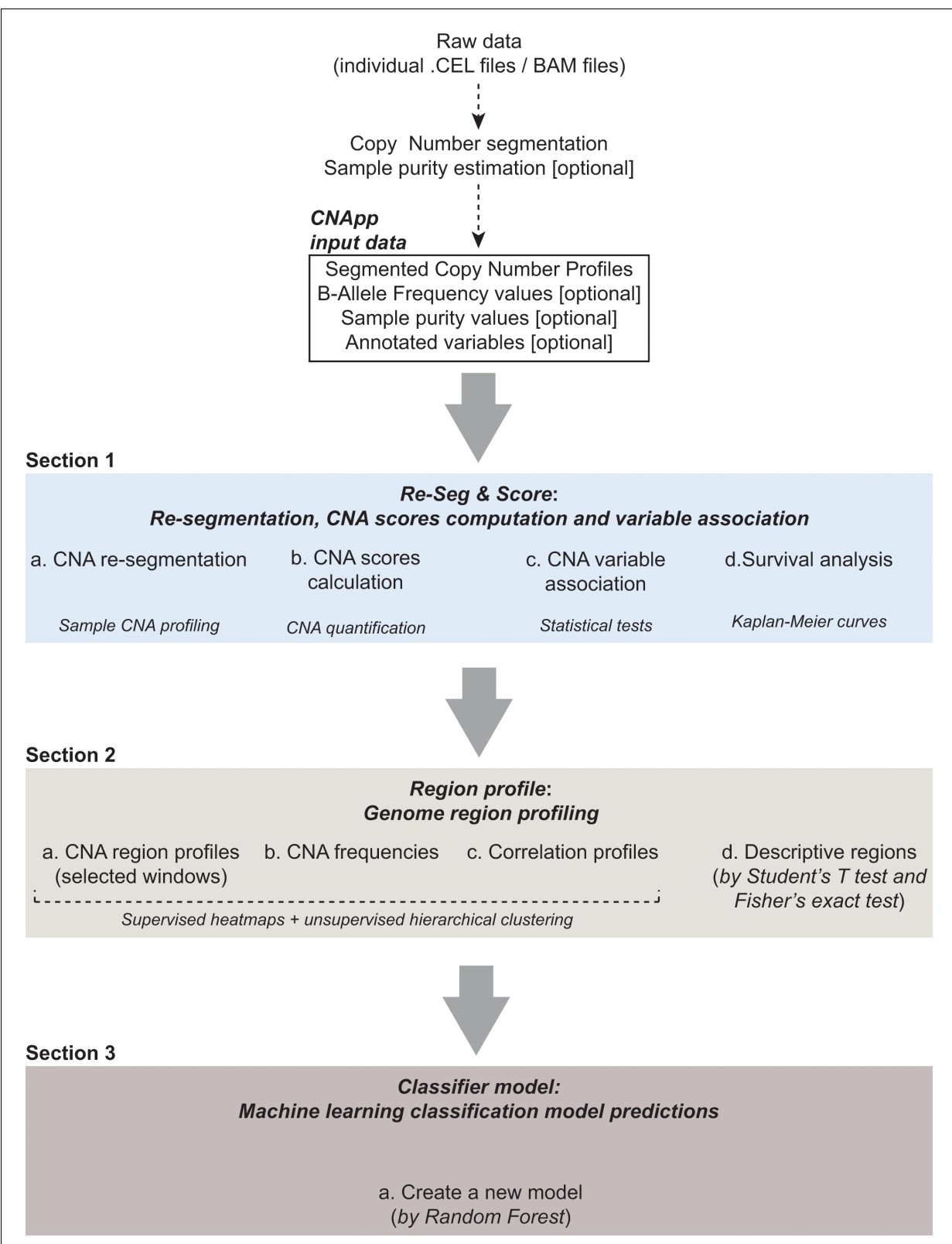

**Figure 1.** CNApp workflow. The diagram depicts the overall processes performed by CNApp and indicates the output for each section. The online version of this article includes the following source data and figure supplement(s) for figure 1:

**Figure supplement 1.** Spearman's rank correlations between CNA scores and fractions of altered genome.

*Figure 1 continued on next page*

*Figure 1 continued*

**Figure supplement 1—source data 1.** Data used for Spearman's rank correlation analyses between CNA scores and fractions of altered genome.

(values ranged from −1.93 to 12.60) highly correlated with the fraction of altered genome affected by both broad and focal CNAs (Spearman's rank correlation for GCS = 0.963 (*Figure 1—figure supplement 1C—source data 1*). Parametric and non-parametric statistical tests are used to establish associations between CNA scores and annotated variables from the input file. Additionally, Kaplan-Meier survival curves are computed using either CNA scores or additional variables.

## Region profile: genome-wide CNA profiling

This section transforms segmented data (either re-segmented data from section one or original segments uploaded by the user) into genomic region profiles to allow sample-to-sample comparisons. Different genomic windows can be selected to compute the genomic profiles (i.e. chromosome arms, half-arms, cytobands, sub-cytobands or 40–1 Mb windows). All segments, or either only broad or only focal can be selected for this analysis. Length-relative means are computed for each window by considering amplitude values from those segments included in each specific window. Default cut-offs for low-level copy number gains and losses (i.e., |0.2|) are used to infer CNA frequencies. Genomic profiles are presented in genome-wide heatmaps to visualize general copy number patterns. Up to six annotation tracks can be added and plotted simultaneously allowing visual comparison and correlation between CNA profiles and different variables, including the CNA scores obtained in section 1. CNA frequency summaries by genomic region and by sample are represented as stacked bar plots. Correlation values and hierarchical clusters are optional.

Importantly, assessing differentially altered regions between sample groups might contribute to discover genomic regions associated with annotated variables and thus unveil the biological significance of specific CNAs. To do so, CNApp interrogates descriptive regions associated with any sample-specific annotation variable provided in the input file. Default statistical significance is set to p-value lower than 0.1. However, p-value thresholds can be defined by the user and adjusted p-value is optional. A heatmap plot allows the visualization and interpretation of which genomic regions are differentially altered between sample groups. By selecting a region of interest, box plots and stacked bar plots are generated comparing *seg.mean* values and alteration counts in Student's t-test and Fisher's test tabs, respectively. Additionally, genes comprised in the selected genomic region are indicated.

## Classifier model: Machine learning classification model predictions

This section allows the user to generate machine learning-based classifier models by choosing a variable to define sample groups and one or multiple classifier variables. To do so, CNApp incorporates the *randomForest* R package (*Liaw and Wiener, 2002*). The model construction is performed 50-times, in which training sets of 75% of samples are used in each permutation and the remaining 25% are classified by applying the trained model. By default, annotation variables from the input file are loaded and can be used either by defining sample groups or as a classifier. If *Re-Seg and Score* and/or *Region profile* sections have been previously completed, the user can upload data from these sections (i.e. CNA scores and genomic regions). Predictions for the model performance are generated and the global accuracy is computed along with sensitivity and specificity by group. Classifier models can be useful to point out candidate clinical or molecular variables to classify sample subgroups. Nevertheless, the output from the *Classifier model* would need to be further validated using alternative prediction algorithms and/or independent datasets. A summary of the data distribution and plots for real and model-predicted groups are visualized. A table with prediction rates throughout the 50-times iteration model and real tags by sample is displayed and can be downloaded.

## Characterization of cancer types based on CNA scores

First, we evaluated the ability of CNApp to analyze and classify cancer types according to CNA scores, and assessed whether CNApp was able to reproduce specific CNA patterns across different cancer types. To do so, by using CNApp default parameters we obtained re-segmented data, CNA scores and cancer-specific CNA profiles for 10,635 tumor samples spanning 33 cancer types from

the TCGA pan-cancer dataset. The distribution of BCS, FCS and GCS confirmed the existence of distinct CNA burdens across cancer types (*Figure 2A—source data 1*). While cancer types such as acute myeloid leukemia (LAML), thyroid carcinoma (THCA) or thymoma (THYM) showed low levels of broad and focal events (GCS median values of −1.67 for LAML, −1.68 for THCA, and −1.52 for THYM), uterine carcinosarcoma (UCS), ovarian cancer (OV) and lung squamous cell carcinoma (LUSC) displayed high levels of both types of genomic imbalances (GCS median values of 2.55, 2.44, and 0.97 for UCS, OV, and LUSC, respectively). Some cancer types displayed a preference for either broad or focal CNAs. For example, kidney chromophobe (KICH) tumors showed the highest levels of broad events (median BCS value of 27), while focal CNAs in this cancer type were very low (median FCS value of 49). In contrast, breast cancer (BRCA) samples displayed high FCS values (median FCS value of 150), while BCS values were only intermediate (median BCS value of 7). Overall correlations between CNA scores were assessed by computing Spearman's rank test, obtaining values of 0.59 between BCS and FCS, 0.90 between BCS and GCS, and 0.85 between FCS and GCS. In addition, we further assessed the correlation between BCS and FCS for each individual BCS value. While tumors with low BCS displayed a positive correlation between broad and focal alterations, tumors did not maintain such correlation in higher BCS values (*Figure 2—figure supplement 1A and B*). This correlation between BCS and FCS is maintained across the 33 cancer types (*Figure 2—figure supplement 1C*).

Subsequent analysis aimed at generating genome-wide patterns for each cancer type based on chromosome-arm genomic windows and the overall corresponding frequencies. In agreement with previous studies (*Beroukhim et al., 2010*), cancer type-specific patterns of genomic gains and losses determined the tissue-of-origin (*Figure 2B*). Additionally, we found that chromosome arms altered in more than 25% across all samples were 1q, 7 p, 7q, 8q and 20q for copy number gains, and 8 p and 17 p for copy number losses. Conversely, chromosome arms affected by CNAs in less than 10% of all cancer types included chromosome arms 2q and 19 p (*Figure 2C*). By using a subset of 20 out of the 33 cancer types for which tumor type information was available, we asked CNApp to compute the average arm-region for each cancer type to assess if they clustered according to their CNA profiles (*Figure 2—figure supplement 2A*). Our analysis showed that correlation values resulting from Pearson's test hierarchically clustered according to the tissue-of-origin from the tumor. Gastrointestinal (colon, rectum, stomach and pancreatic), gynecological (ovarian and uterine) and squamous (cervical, head and neck, and lung) cancers clustered together based on specific CNA profiles for each group (*Figure 2D*). Intriguingly, correlation profiles using 5 Mb windows and only considering focal alterations showed a very similar degree of clustering based on the tissue of origin (*Figure 2—figure supplement 2B&C*).

## Identification of recurrent CNAs in hepatocellular carcinoma: Benchmark with other available tools

Next, we attempted to test the ability of CNApp to identify recurrent broad and focal CNAs in a large cohort, and to assess the impact of the customizable parameters to describe CNA profiles. For that reason, we chose to perform CNA analysis of 370 samples from TCGA corresponding to the Liver Hepatocellular Carcinoma (LIHC) cohort. The pattern of recurrent broad and focal CNAs identified by GISTIC2.0 in the TCGA study (*Ally et al., 2017*) was similar to earlier reports, confirming the suitability of this cohort and the consistent identification of a CNA profile for hepatocellular carcinoma (HCC) (*Chiang et al., 2008*; *Guichard et al., 2012*; *Schulze et al., 2015*; *Totoki et al., 2014*; *Wang et al., 2013*).

By applying the default parameters of CNApp to the LIHC dataset and selecting chromosome arms as genomic regions to assess broad events, we consistently found copy number gains at 1q (56%) and 8q (46%), and copy number losses at 8 p (62%) and 17 p (47%) as the most frequent alterations (*Figure 3A*). These CNAs are the same as those identified by GISTIC2.0; however, frequencies were slightly lower (*Supplementary file 1*). Similarly, GISTIC2.0 detected significant gains with rates between 25–40% on eight additional chromosome-arms, including 5 p, 5q, 6 p, 20 p, 20q, 7 p, 7q, and 17q, which were also identified by CNApp, but in 20–30% of the samples. Likewise, GISTIC2.0 detected significant broad deletions at a frequency between 20% and 40% on 18 additional chromosome-arms, of which 4q, 6q, 9 p, 13q, 16 p, and 16q losses were observed at frequencies $\geq$ 20% by CNApp, and the rest of them displayed rates between 10% and 20%. Therefore, the identification of CNAs in CNApp is very consistent with those described by GISTIC2.0. Differences in frequencies

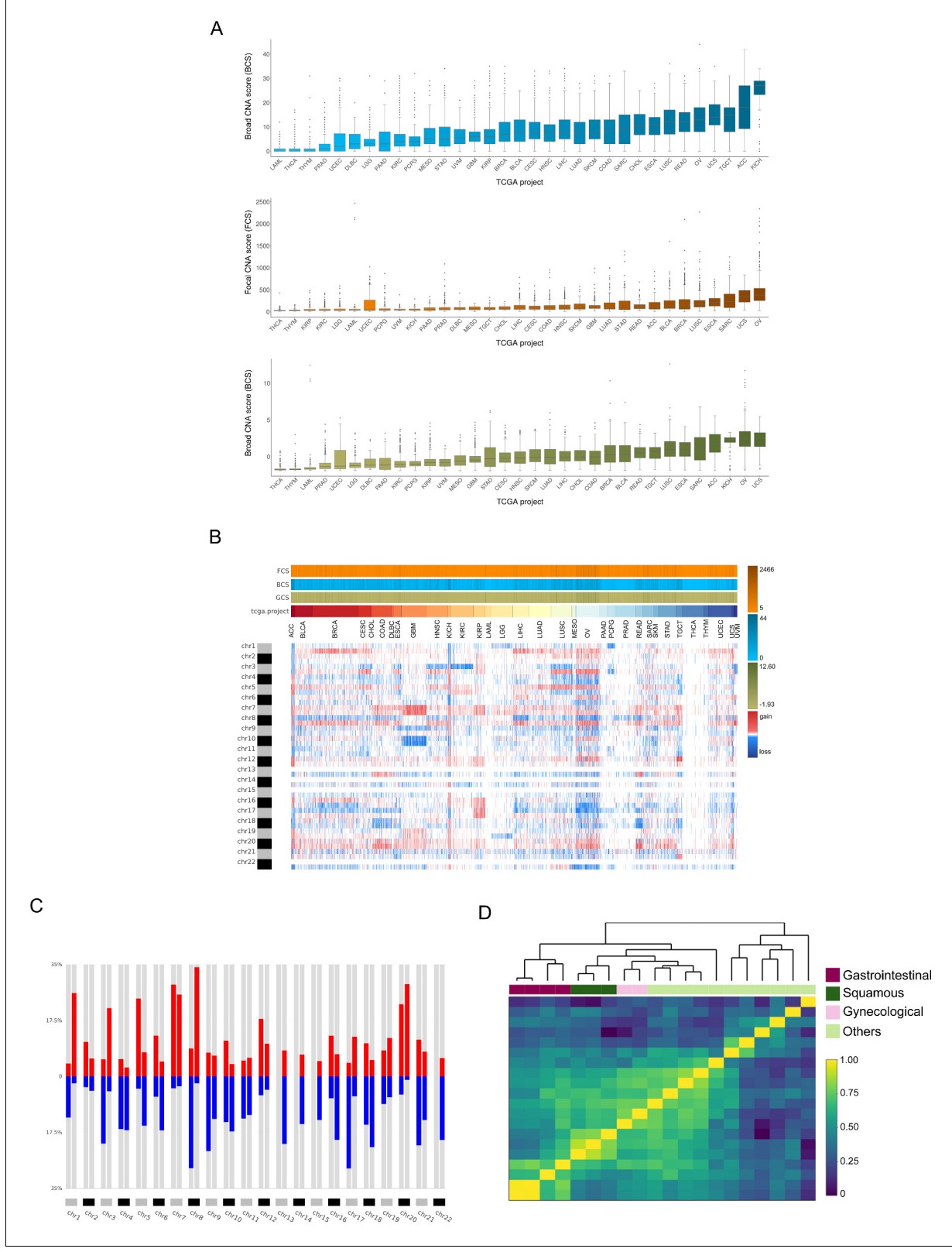

**Figure 2.** Analysis of the TCGA pan-cancer dataset and clustering by tumor type. CNApp outputs to characterize pan-cancer 10,635 samples including 33 TCGA cancer types. (**A**) Broad, Focal and Global CNA scores (BCS, FCS and GCS, respectively) distribution across the 33 cancer types (*Figure 2— source data 1*). (**B**) Genome-wide chromosome arm CNA profile heatmap for 10,635 samples considering broad and focal events. Annotation tracks for FCS, BCS and GCS are presented. (**C**) Arm regions frequencies as percentages relative to the TCGA pan-cancer dataset (red for gains and blue for

*Figure 2 continued on next page*

*Figure 2 continued*

losses). (**D**) Heatmap plot showing 20 out of the 33 TCGA cancer type profile correlations, by Pearson's method, hierarchically clustered by tissue of origin. Gastrointestinal, gynecological and squamous cancers are clustering consistently in their respective groups.

The online version of this article includes the following source data and figure supplement(s) for figure 2:

**Source data 1.** Broad, Focal and Global CNA scores across the pan-cancer cohort.
**Figure supplement 1.** Correlation plots between FCS and BCS.
**Figure supplement 2.** Clustering analysis between cancer types.

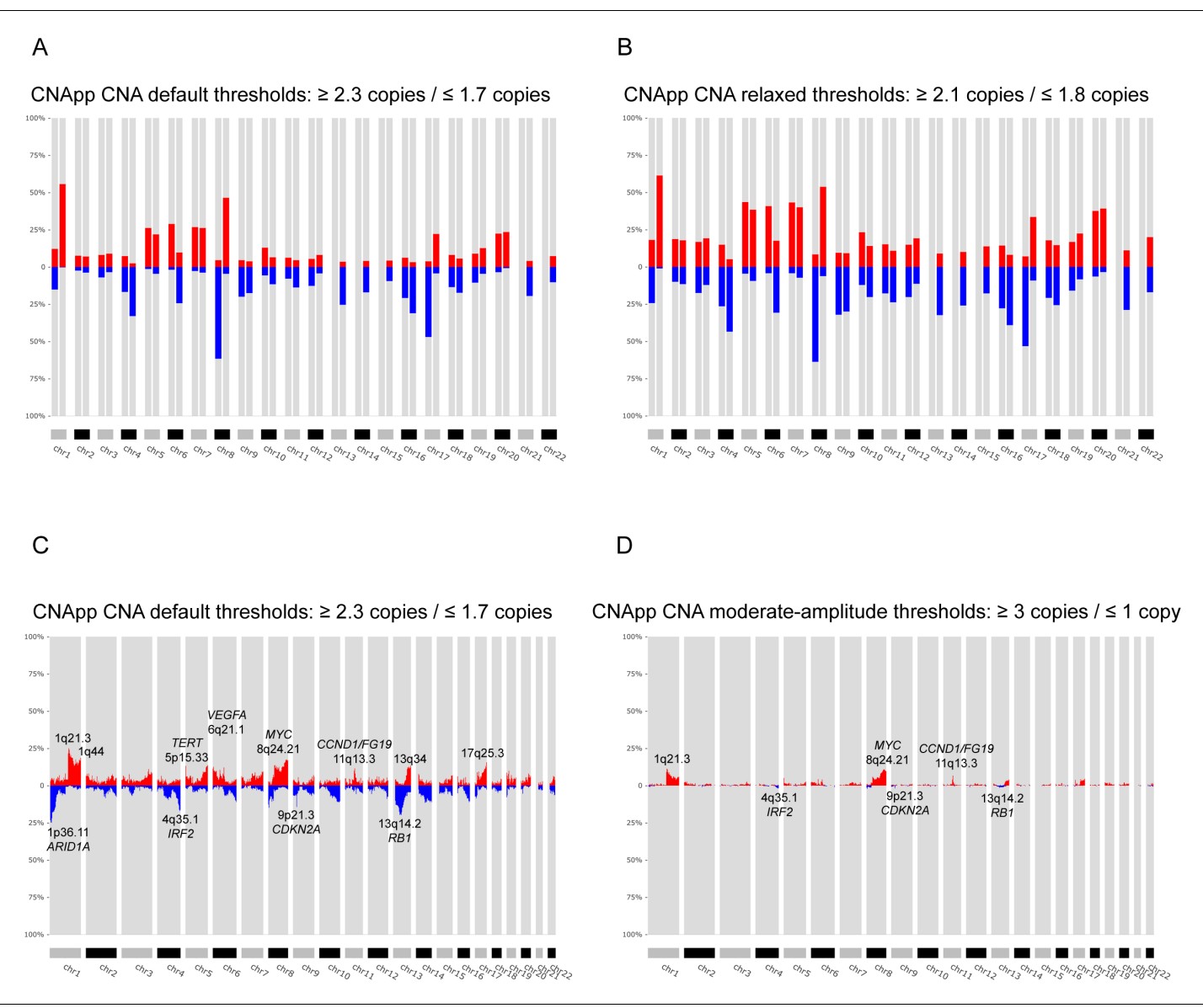

**Figure 3.** Identification of recurrent broad and focal CNAs. Calculation of broad and focal CNA frequencies using several parameters in CNApp in order to describe the genomic landscape of LIHC. (**A**) CNApp frequencies for chromosome arm regions using default cutoffs, corresponding to 2.3/1.7 copies for gains and losses, respectively. (**B**) CNApp frequencies for chromosome arm regions relaxing cutoffs to make them equivalent to those of GISTIC2.0. (**C**) CNApp frequencies of focal events using default thresholds and sub-cytobands genomic regions. (**D**) Frequencies of focal events from moderate- to high-amplitude levels using sub-cytobands genomic regions.

The online version of this article includes the following figure supplement(s) for figure 3:

**Figure supplement 1.** Genomic profiles corresponding to the LIHC dataset using alternative thresholds.

might be expected due to the lower copy number amplitude thresholds used by GISTIC2.0 in comparison with CNApp default cutoffs (|0.1| vs. |0.2|, corresponding to ~2.14/1.8 copies vs. 2.3/1.7 copies, respectively). Indeed, previous reports analyzing CNAs in other HCC cohorts and using greater copy number thresholds, showed frequencies of alterations much more similar to those estimated by CNApp (*Chiang et al., 2008*; *Guichard et al., 2012*; *Schulze et al., 2015*; *Wang et al., 2013*). To assess the impact of modifying CNApp amplitude thresholds, we analyze the same dataset dropping the cutoff to |0.1|. By doing so, the overall number of broad CNAs increased, reaching frequency values similar or even higher than those reported by GISTIC2.0 (*Figure 3B* and *Supplementary file 1*). Of note, such drop from |0.2| to |0.1| might result in the identification of subclonal genomic imbalances, which are very frequent in tumor samples (*McGranahan and Swanton, 2017*), and it would also be of utility to compensate for low tumor purities. To evaluate the impact of other customizable parameters of CNApp to the results, we also tested whether the identification of broad events was affected by: (i) the relative length to classify a segment as *arm-level* alteration, and (ii) the re-segmentation provided by CNApp. As expected, increasing the percentage of chromosome arm required to classify a CNA segment as *arm-level* (from ≥50% to ≥70%) or skipping the re-segmentation step led to an underestimation of some broad events, whereas decreasing the percentage of chromosome arm (from ≥50% to ≥40%) resulted in the opposite (*Figure 3—figure supplement 1A–C* and *Supplementary file 1*).

As far as focal CNAs are concerned, CNApp and GISTIC2.0 use different strategies to quantify their recurrence. Therefore, the comparison between the two methods was evaluated in a more indirect manner. GISTIC2.0 generates minimal common regions (also known as 'peaks') that are likely to be altered at high frequencies in the cohort, which are scored using a Q-value and may present a wide variety of genomic lengths (*Mermel et al., 2011*). Instead, CNApp allows dividing the genome in windows of different sizes, calculating average copy number amplitudes for all segments included within each window. We reasoned that considering the length of GISTIC2.0 reported 'peaks', CNApp might also be capable of identifying recurrent focal altered regions by dividing the genome in smaller windows. To test our hypothesis, we asked CNApp to calculate the frequency of focal gains and losses by dividing the genome by sub-cytobands. As a result, CNApp consistently localized the most frequently altered sub-cytobands, including gains at 1q21.3 (25%), 8q24.21 (17%, *MYC*), 5p15.33 (13%, *TERT*), 11q13.3 (12%, *CCND1/FGF19*) and 6p21.1 (11%, *VEGFA*), and losses at 13q14.2 (20%, *RB1*), 1p36.11 (18%, *ARID1A*), 4q35.1 (17%, *IRF2*) and 9p21.3 (14%, *CDKN2A*), which are in agreement with previous studies in HCC (*Figure 3C* and *Supplementary file 2*) (*Chiang et al., 2008*; *Guichard et al., 2012*; *Schulze et al., 2015*; *Wang et al., 2013*). Compared to GISTIC2.0, CNApp reported 14 of the 27 significant amplifications and 14 of the 34 significant deletions at rates > 10%, and the remaining alterations displaying rates between 4–10% (*Supplementary file 3*) (*Wang et al., 2013*). Most importantly, regions with the highest frequency detected by CNApp showed a good match with lowest GISTIC2.0 Q-residual values, indicating that the most significant 'peaks' identified by GISTIC2.0 were actually included in the most recurrently altered sub-cytobands reported by CNApp.

Recurrent focal alterations occur at lower frequencies than broad events (*Beroukhim et al., 2010*). Previous studies describing the genomic landscape of HCC mostly focused on focal high-amplitude CNAs (>3 copies for gains and <1.3 copies for losses), thus reporting lower frequencies than those estimated by CNApp using default thresholds (*Chiang et al., 2008*; *Guichard et al., 2012*; *Schulze et al., 2015*). In our analysis, excluding the low-level alterations and evaluating only the moderate and high-amplitude events (≥3 and ≤1 copies), amplifications reached maximum rates of 11%, whereas high-level losses only reached ~2% (*Figure 3D* and *Supplementary file 2*). Top recurrent focal gains involved sub-cytobands 1q21.3 (11%), 8q24.21 (11%, *MYC*), 11q13.3 (7%, *CCND1/FGF19*), and 5p15.33 (5%, *TERT*). Recurrent losses estimated at ~2% of the samples included 13q14.2 (*RB1*), 9p21.3 (*CDKN2A*), 4q35.1 (*IRF2*), and 8p23.1. Although slight discrepancies between frequencies might be explained by minimal variability in the copy number threshold, CNApp results are highly consistent with previous reports (*Chiang et al., 2008*; *Guichard et al., 2012*; *Schulze et al., 2015*).

In order to assess genomic differences between independent sample sets, CNApp determines significance based on regression analysis and statistical tests such as Student's t-test or Fisher's exact test, which allow association of genomic regions based on the *seg.mean* value or the presence of alterations with specific samples, respectively. To test the suitability of our method, we analyzed a

subset of 100 randomly selected samples from the LIHC (n = 50) and COAD (n = 50) TCGA cohorts and we compared our results with those obtained by the recently published tool CoNVaQ (*Larsen et al., 2018*). Fourteen CNAs (5 gains and 9 losses) showing significant differences between COAD and LIHC were identified by CoNVaQ with a *Q* < 0.05. The lengths of these CNAs ranged from 38 to 133 Mb, with an average of 67 Mb, suggesting that they were mainly broad events. Consequently, in CNApp, we selected genomic windows corresponding to chromosome arms, and used the default CNA cut-offs (i.e. |0.2|) and maximum number of base pairs allowed for merging adjacent segments (i.e. 1,000,000 bp) for comparison. By doing so, 9 out of 14 events identified by CoNVaQ were also detected by CNApp with an adjusted p<0.05 (Fisher's exact test). For the remaining five events, four of them were found with an adjusted p=0.06–0.1, thus suggesting a good correlation between CNApp and CoNVaQ when considering broad CNAs.

## Classification of colon cancer according to CNA scores and genomic regions

A proposed taxonomy of colorectal cancer (CRC) includes four consensus molecular subtypes (CMS), mainly based on differences in gene expression signatures (*Guinney et al., 2015*). Briefly, CMS1 includes the majority of hypermutated tumors showing microsatellite instability (MSI), high CpG island methylator phenotype (CIMP), and low levels of CNAs; CMS2 and CMS4 typically comprise microsatellite stable (MSS) tumors with high levels of CNAs; and finally, mixed MSI status and low levels of CNAs and CIMP are associated with CMS3 tumors. A representative cohort of 309 colon cancers from the TCGA Colon Adenocarcinoma (COAD) cohort (*Cancer et al., 2012*) with known CMS classification (CMS1, N = 64; CMS2 N = 112; CMS3 N = 51; CMS4 N = 82) and MSI status (MSI, N = 72; MSS, N = 225) was analyzed by using CNApp. In agreement with Guinney and colleagues, survival curves generated by CNApp indicated that CMS1 patients after relapse showed the worst survival rates as compared to CMS2 patients (*Figure 4—figure supplement 1A*) (*Guinney et al., 2015*). Next, we asked CNApp to perform the re-segmentation step using the default copy number thresholds and excluding segments smaller than 500 Kbp to avoid technical background noise. Then, broad CNAs were considered to generate genomic region profiles using chromosome-arm windows. As expected, the CNA frequency plot displayed the most commonly altered genomic regions in sporadic CRC (*Figure 4—figure supplement 1B*) (*Camps et al., 2008*; *Cancer et al., 2012*; *Meijer et al., 1998*; *Nakao et al., 2004*; *Ried et al., 1996*). Most frequently altered chromosome arms included gains of 7 p, 7q, 8q, 13q, 20 p, and 20q, and losses of 8 p, 17 p, 18 p, and 18q, occurring in more than 30% of the samples (*Figure 4A*). On the other hand, focal CNA patterns were obtained by generating genomic profiles by sub-cytobands. Of note, five out of six losses and five out of 18 gains were also identified by GISTIC2.0 in the COAD TCGA cohort (*Cancer et al., 2012*).

Subsequently, we performed integrative analysis of genomic imbalances, CMS groups, and CNA scores. By using CNApp, we assessed whether CNA scores were able to classify colon cancer samples according to their CMS. While BCS established significant differences between CMS paired comparisons (p≤0.0001, Student's t-test), FCS poorly discerned CMS1 from three and CMS2 from 4 (*Figure 4B—source data 1* and *Figure 4—figure supplement 1C—source data 1*). Thus, we reasoned that broad CNAs rather than focal were able to better discriminate between different CMS groups. In fact, the distribution of CMS groups based on BCS resembled the distribution of somatic CNA counts defined by GISTIC2.0 (*Guinney et al., 2015*).

Next, we integrated the BCS and the CMS groups with the microsatellite status. Our results showed an average BCS of 1.51 ± 2.11 and 10.25 ± 5.92 for MSI (N = 72) and MSS (N = 225) tumors, respectively. By applying CNApp's *Classifier model* to the COAD cohort, we chose MSI and MSS status (72 MSI and 225 MSS samples). Global accuracy in average for the 50-permutation in the RandomForest model was 82.2%. Re-classification of MSI and MSS samples was successfully achieved and distribution of BCS values across MSI and MSS-predicted samples is plotted in *Figure 4—figure supplement 1D*. Intersection between predicted MSI and MSS groups resulted in a BCS value of 4.75, and further analysis by ROC curve implementation between MSI and MSS groups with BCS values as classifier proxy resulted in an AUC of 0.917. In addition, the intersection between sensitivity and specificity from ROC analysis resulted in a BCS value of 3.5 (*Figure 4—figure supplement 1E*). Altogether, we decided to implement a BCS value of 4 as a threshold to re-classify samples according to their MSI status. In order to validate that a BCS value of 4 was able to predict microsatellite

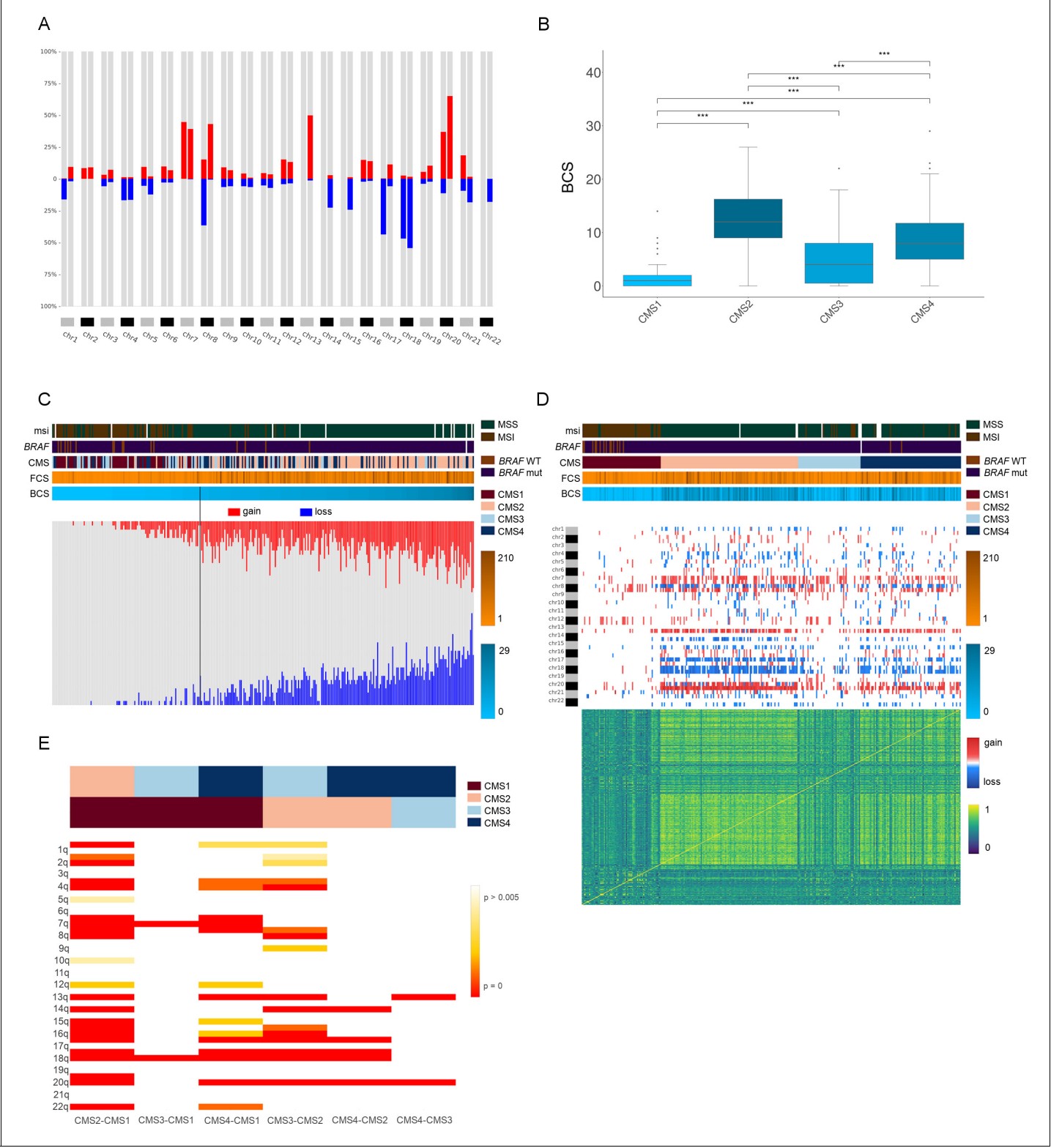

**Figure 4.** Genomic characterization of colon cancer according to the CMS classification. (**A**) Arm-region frequencies of 309 colon cancer samples using CNApp default thresholds for CNAs. (**B**) BCS distribution by CMS sample groups (*Figure 4—source data 1*). FCS distribution by CMS sample groups is presented in *Figure 4—figure supplement 1C*. Wilcoxon rank-sum test significance is shown as p-value≤0.001 (***); p-value≤0.01 (**); p-value≤0.05 (*); p-value>0.05 (ns). (**C**) Number of gained and lost chromosome arms for each sample distributed according to the BCS values. Note that a cutoff at four is indicated with a black line. Annotation tracks for microsatellite instability (*msi*), *BRAF* mutated samples (*braf_mut*), CMS groups (*cms_label*), FCS

*Figure 4 continued on next page*

*Figure 4 continued*

and BCS are displayed. (D) Genome-wide profiling by chromosome arms distributed according to the CMS group. Annotation tracks for microsatellite instability (msi), *BRAF* mutated samples (braf_mut), CMS groups (cms_label), FCS and BCS are displayed. Sample-to-sample correlation heatmap plot by Pearson's method is shown below. (E) Differentially altered chromosome arm regions between CMS groups (CMS1, CMS2, CMS3 and CMS4). Heatmap plot displaying the significance between CMS groups paired comparisons. Student's T-test was applied and multiple testing correction by BH method was used to assess differences in chromosome arm values between groups. Adjusted p-values are displayed.

The online version of this article includes the following source data and figure supplement(s) for figure 4:

**Source data 1.** BCS by CMS sample group.
**Figure supplement 1.** Genomic analysis of the COAD cohort.
**Figure supplement 1—source data 1.** FCS by CMS sample group.

status, we used a completely independent CRC cohort (*Berg et al., 2019*). The validation dataset consisted of segmented data from 147 samples with known microsatellite status and CMS annotation (MSI, N = 28; MSS, N = 119; CMS1, N = 27; CMS2, N = 61; CMS3, N = 29; CMS4, N = 30). We then ran CNApp to obtain BCS values for each sample and applied our BCS of 4 as the threshold to re-classify samples (i.e. MSI if BCS ≤4, and MSS if BCS >4) using the *Classifier model*. By doing so, we obtained a global accuracy of 81%, which is similar to the accuracy obtained in the COAD dataset (82.2%), thus suggesting that a BCS equal to four might be considered as a universal threshold to determine microsatellite status in CRC.

Applying the cutoff of 4 to the COAD cohort, 186 out of 225 (83%) of MSS tumors showed BCS values greater than 4 (*Figure 4C*). In contrast, 39 (17%) MSS tumors showed a BCS of 4 or lower, corresponding to three CMS1, six CMS2, 18 CMS3 and 12 CMS4 tumors, further demonstrating the existence of MSS tumors with a very low CNA burden. On the other hand, seven MSI tumors showed BCS higher than 4. Among them, five samples displayed genomic imbalances typically associated with the CRC canonical pathway, including a focal amplification of *MYC*, unveiling tumors with co-occurrence of MSI and extensive genomic alterations (*Trautmann et al., 2006*). Our dataset comprised nine out of 51 CMS3 tumors with MSI. Intriguingly, two of them showed focal deletions on chromosome two involving *MSH2* and *MSH6*, suggesting the inactivation of these mismatch repair genes through a focal genomic imbalance. In fact, 46% of CMS3 MSS tumors showed BCS below 4, in agreement with the finding that CMS3 tumors display low levels of somatic CNAs.

Moreover, CNApp enabled the identification of possible sample misclassifications by integrating CMS annotation and *BRAF*-mutated sample status. As expected, CMS1 cases were enriched for *BRAF* mutation, although two CMS4 samples also showed mutations in *BRAF*. One of these samples showed a BCS of 11, displaying canonical CNAs. In contrast, the other CMS4 *BRAF*-mutated sample showed MSI and a BCS of 0, similar features as CMS1. Likewise, four *BRAF*-wt samples, classified within the CMS4 group, displayed MSI and a BCS of 0, thus being candidates to be labeled as CMS1 based on the levels of CNAs (*Figure 4D*). These disparities are of utmost importance since recent studies reported that high copy number alterations correlate with reduced response to immunotherapy (*Davoli et al., 2017*). Importantly, it has been suggested that MSI status might be predictive of positive immune checkpoint blockade response in advanced CRC, probably due to the low levels of CNA usually presented by MSI tumors (*Le et al., 2015*).

We then asked CNApp to compare differentially represented genomic regions between all CMS groups based on a Student's t-test or Fisher's test with adjusted p-value. By applying a Student's t-test, we observed that CMS1 resembled CMS3, except for the gain of chromosome seven and the loss of 18q, which were regions commonly altered in CMS3 samples with BCS above 4 (adjusted p≤0.001, Student's t-test) (*Figure 4E*). Even though only subtle CNA differences between CMS2 and CMS4 were identified, the loss of 14q was significantly more detected in CMS2 (42%) than in CMS4 (17.1%) (adjusted p≤0.005, Student's t-test). The gain of 12q was more frequently associated with CMS1 than CMS2 (adjusted p≤0.005, Student's t-test), in agreement with previous studies reporting that the gain of chromosome 12 is associated with microsatellite unstable tumors (*Figure 4E*) (*Trautmann et al., 2006*). Intriguingly, the gain of the chromosome arm 20q alone mimicked the distribution of somatic CNAs defined by GISTIC2.0 across consensus subtype samples (*Figure 4—figure supplement 1F*) (*Guinney et al., 2015*).

Finally, applying machine learning-based prediction models to classify samples by the most discriminative descriptive regions across CMS groups (i.e. 13q, 17 p, 18, and 20q), CNApp reached

55% of accuracy to correctly predict CMS. In fact, the occurrence of these genomic alterations was able to differentiate CMS2 from CMS4 with an accuracy of 70%, and CMS1 from CMS3 with a 72.3% accuracy. As expected, this set of genomic alterations distinguished CMS1 from CMS2 samples with an accuracy of 95%.

## Discussion

Here, we present CNApp, a web-based computational tool that provides a unique framework to comprehensively analyze and integrate CNAs associated with molecular and clinical variables, assisting data-driven research in the biomedical context. Although CNApp has been developed using segmented genomic copy number data obtained from SNP-arrays, the software is also able to accommodate segmented data from any next-generation sequencing platform.

CNApp transforms segmented data into genomic profiles, allowing sample-by-sample comparison and the assessment of differentially altered genomic regions, which can then be selected by the user to assess classifier variables by computing machine learning-based models. Importantly, besides identifying the impact of specific CNAs, CNApp provides the unique opportunity to establish associations between the burden of genomic alterations and any clinical or molecular variable. To do so, CNApp calculates CNA scores, a quantification of the broad (BCS), focal (FCS) and global (GCS) levels of genomic imbalances for each individual sample. The fact that high levels of aneuploidy may correlate with tumor immune evasion markers and that CNA burdens may be an independent prognostic factor for cancer-specific lethality in some cancers exemplifies the potential association of CNA scores with clinical outcomes (*Buccitelli et al., 2017*; *Davoli et al., 2017*; *Hieronymus et al., 2018*; *Taylor et al., 2018*). To note, CNA scores are calculated after an optional process of re-segmentation that enables to redefine CNA boundaries and to adjust sample-specific copy number thresholds by correcting for tumor purity estimates.

In agreement with recently reported findings (*Beroukhim et al., 2010*; *Hoadley et al., 2018*; *Taylor et al., 2018*), CNApp was benchmarked by analyzing 10,635 samples spanning 33 cancer types from the TCGA pan-cancer dataset, and was able to cluster major tumor types according to CNA patterns. Moreover, the software successfully reproduced the well-characterized genomic profile of HCC and CRC, considering both broad and focal events, demonstrating the reliability of CNApp in identifying regions encompassing the most recurrent CNAs (*Ally et al., 2017*; *Cancer et al., 2012*).

Finally, applying CNApp to the TCGA colon cancer sample set, for which MSI status and CMS classification was well annotated, we determined that a BCS value of 4 discriminates MSI from MSS tumors with high accuracy, reinforcing the utmost significance of quantifying the CNA burdens. Most importantly, due to the inverse correlation between MSI and aneuploidy in CRC, our results suggest that this BCS value could be established as a cutoff to define the edge between low and high aneuploid tumors. In fact, while high aneuploid tumors show poor response to immunotherapy, it has been suggested that CMS1 microsatellite unstable tumors are likely to show a positive response to immune checkpoints inhibitors (*Kalyan et al., 2018*; *Le et al., 2015*). However, BCS was not associated with overall survival in patients after relapse (data not shown). Moreover, specific genomic regions defined by CNApp contributed to classify the CMS groups, confirming the functional importance of specific genomic imbalances in the pathogenesis of this disease and providing insights into the classification of CRC based on CNA profiles.

In summary, although our results ought to be further validated in independent cohorts, here we show that CNApp enables not only the fundamental analysis of CNA profiles, but also the functional understanding of CNAs in the context of clinical outcome and their potential use as biomarkers, thus becoming an asset to the cancer genomics community.

## Materials and methods

### Data availability

#### Pan-cancer cohort and clinical annotation

Affymetrix SNP6.0 array copy number segmented data (Level 3) from 10,635 samples spanning 33 cancer types from TCGA pan-cancer dataset were downloaded from the Genomic Data Commons

Data Portal (GDC Data Portal, RRID:SCR_014514) (*Grossman et al., 2016*). This dataset included the 370 Liver Cancer-Hepatocellular Carcinoma (LIHC) samples used for the analysis of recurrent CNAs and the subset of 309 samples from Colon Adenocarcinoma (COAD) for which the colorectal cancer consensus molecular subtype (CMS) was known (*Guinney et al., 2015*).

Clinical annotation for the 309 COAD samples was retrieved by using *TCGAbiolinks* R package (TCGAbiolinks, RRID:SCR_017683) in order to extract survival information for each sample (*Colaprico et al., 2016*).

## GISTIC data from TCGA: LIHC cohort

GISTIC 2.0.22 (*Ally et al., 2017*) copy number results from 370 LIHC samples included in the TCGA (The Cancer Genome Atlas, RRID:SCR_003193), were downloaded from the Broad Institute GDAC Firehose (https://gdac.broadinstitute.org/). Parameters used for the analysis are detailed in the same GDAC repository. Specifically, parameters conditioning the definition of the CNAs and of interest for our comparison were publicly reported with the following values: *amplification* and *deletion thresholds*: 0.1; *broad length cutoff*: 0.7; *joint segment size*: 4.

## Software and tool availability

CNApp can be accessed at https://tools.idibaps.org/CNApp/. It was developed using Shiny R package (version 1.1.0), from R-Studio (Shiny, RRID:SCR_001626) (*Chang et al., 2018*). The tool was applied and benchmarked while using R version 3.4.2 (2017-09-28) – 'Short Summer'. List of packages, libraries and base coded are freely available at GitHub, and instructions for local installation are also specified.

## Baseline adjustment of seg.mean values by sample purity

The amount of non-aberrant cell admixture may differ between cancer samples, necessitating separate adjustment of copy number thresholds for each assayed sample. To homogenize the analysis, purity estimations are used by CNApp to apply a baseline adjustment of the *seg.means* before the subsequent CNA calling by CNA amplitude thresholds. This adjustment allows for subsequent sample-to-sample comparison. *Seg.mean* values (*n*) by sample (*x*), when purity (*r*) available, are re-computed into new *seg.mean* values (*N*) as follows:

$$N(x) = 2^{n(x)+(r(x)-1)}$$

When purity is not provided, 100% purity (r = 1) is assumed. Minimal purity accepted is 40% (r = 0.4), setting to 40% all purities below. Threshold of 40% also sets the negative capping value (C) as the maximal loss possible when heterozygous deletion happens (from 2 copies to one copy):

$$C = log_2\left(\frac{1}{2} \cdot 0.4\right) = -2.32$$

## CNA scores computation

Segments resulting from re-segmentation (or original segments from input file when re-segmentation is skipped) are classified in *chromosomal*, *arm-level* and *focal* events by considering the relative length of each segment to the whole-chromosome or chromosome arm. Using default parameters, segments are tagged as *chromosomal* when 90% or more of the chromosome is affected; as *arm-level* when 50% or more of the chromosome arm is affected; and as *focal* when affecting less than 50% of the chromosome arm. Percentages for relative lengths are customizable.

Broad (chromosomal and arm-level) and focal alterations are then weighted according to their amplitude values (*seg.mean*) and taking into account copy number amplitude ranges defined by CNA calling thresholds. Default amplitude thresholds to define these ranges, CNA levels and their corresponding absolute copy number are presented in *Table 1*.

In order to take into account the relative chromosome length from each segment when FCS is computed, coverage punctuation for focal events is implemented as specified in *Table 2*.

*Broad CNA Score* (BCS): for a total *N* of broad events in a sample (*x*), it equals to the summation of segments weights (*A*) in that corresponding sample and being *i* the corresponding segment:

**Table 1.** Corresponding weights (*A*) for each threshold, CNA level and its absolute copy number.

| Thresholds (Log2ratio values) | CNA level | Copy number | Weights (*A*) |
|---|---|---|---|
| 1 | High-level gain | ≥4 copies | 3 |
| 0.58 | Medium-level gain | [3–4) copies | 2 |
| 0.2 | Low-level gain | [2.3–3) copies | 1 |
| −0.2 | Low-level loss | (1–1.7] copies | 1 |
| -1 | Medium-level loss | (0.6–1] copies | 2 |
| −1.74 | High-level loss | ≤0.6 copies | 3 |
| [−0.2–0.2] | Copy-neutral loss-of-heterozygosity (CN-LOH) | [1.7–2.3] copies BAF ≥ 0.25 | 2 |

$$BCS(x) = \sum_{i=1}^{N} A_i$$

*Focal CNA Score* (FCS): same as in BCS, with an additional pondering value *L* included to the summation, which captures the relative size of the chromosome-arm coverage of each focal CNA):

$$FCS(x) = \sum_{i=1}^{N} A_i \cdot L_i$$

*Global CNA Score* (GCS): for a sample *x*, it is calculated as the summation of normalized BCS and FCS values, where *meanBCS* and *meanFCS* stand for mean values of BCS and FCS from total samples, respectively, and *sdBCS* and *sdFCS* stand for standard deviation values of BCS and FCS from total samples, respectively:

$$normBCS(x) = \frac{BCS(x) - meanBCS}{sdBCS} \qquad normFCS(x) = \frac{FCS(x) - meanFCS}{sdFCS}$$

$$GCS(x) = normBCS(x) + normFCS(x)$$

## Correlation between CNA scores and the fraction of genome altered

Correlation between broad, focal and global CNA scores (BCS, FCS and GCS, respectively) was assessed by computing Spearman's rho statistic. The TCGA pan-cancer dataset including 10,635 samples spanning 33 cancer types was used to perform these associations. This data was downloaded from the Genomic Data Commons Data Portal (GDC Data Portal, RRID:SCR_014514) (*Grossman et al., 2016*).

To evaluate the applicability of the CNA scores when it comes to sample CNA level assessment, we performed correlation analyses between CNA scores and the altered genome fraction. BCS and FCS were specifically correlated to the altered genome fraction by broad (i.e., chromosomal and arm-level events) and focal alterations, respectively. To do so, we applied the re-segmentation procedure and CNA scores computation on a TCGA dataset comprised by 10,635 primary tumors from 33 TCGA projects analyzed by Affymetrix 6.0 SNP-array and DNACopy (DNACopy, RRID:SCR_012560) (*Venkatraman and Olshen, 2007*). Global altered genome fraction (*altFract*) was computed,

**Table 2.** Coverage punctuation (*L*) for focal events.

| % chromosome-arm coverage | Coverage punctuation (*L*) |
|---|---|
| ≤5% | 1 |
| >5% to≤15% | 2 |
| >15% to≤30% | 3 |
| >30% | 4 |

for each sample (*x*), by the summation of all copy number events lengths (*l*) and divided by total length of human genome (*hgLength*).

$$altFract(x) = \frac{\sum_{i=1}^{N} l_i}{hgLength}$$

Altered genome fractions for broad and focal events were computed, for each sample (*x*), by the summation of broad and focal alterations lengths (*l*), respectively, and divided by total length of human genome (*hgLength*).

$$Broad\ altFract\ (x) = \frac{\sum_{i=1}^{N} l_{i(Broad)}}{hgLength}$$

$$Focal\ altFract\ (x) = \frac{\sum_{i=1}^{N} l_{i(Focal)}}{hgLength}$$

Correlation tests were performed by applying Spearman's rho statistic to estimate a rank-based measure of association.

## Associations between CNA scores and annotated variables

Annotated variables from input file are statistically associated with CNA scores computed by CNApp in *Re-Seg and Score* section. Different association tests are applied according to variable class (i.e. categoric or numerical), as described in *Table 3*. Both parametric and non-parametric tests are computed to present p-values in order to assess statistical significance for each association.

## Survival analysis

Survival analysis by Kaplan-Meier was performed using *survival* and *survminer* R packages (CRAN, RRID:SCR_003005).

## Genomic region profiles computation

*Region profiling* section allows genome segmentation analysis by user-selected windows (i.e. arms, half-arms, cytobands, sub-cytobands, and 40 Mb till 1 Mb). In order to do that, windows files were generated for each option and genome build (*hg19* and *hg38*). Cytobands file *cytoBand.txt* from UCSC page and for both genome builds was used as mold to compute regions (*Tyner et al., 2017*).

Segmented samples are transformed into genome region profiles using genomic windows selected by user. Segments from each sample are consulted to assess whether or not overlap with the window region. Thus, window-means (*W*) are computed for each genomic window by collecting segments (*t*) overlapping with window-region (*i*). Segments with *loc.start* or *loc.end* position falling within the region are collected, as well as those segments embedding the entire region. At this point, the summation of each segment-mean (*S*) corrected by the relative window-length (*L*) affected by the segment length (*l*) is performed:

$$W(i) = \sum_{t=1}^{n} S_t \cdot \frac{l_t}{L(i)}$$

## Descriptive regions assessment

Potential descriptive regions between groups defined by the annotated variables provided in the input file can be studied and p-values are presented to evaluate significance in differentially altered regions between those groups. The alterations can be considered as (1) numerical continuous (*seg. mean* values) and (2) categorical variables (gains, losses and non-altered). In the first case, to assess statistical significance between groups Student's T-test is applied, whereas in the second situation the significance is assessed by applying the Fisher's exact test. False discovery rate (FDR) adjustment is performed using the Benjamini-Hochberg (BH) procedure in both cases and corrected p-values (*Adj.p-value*) or non-corrected p-values (*p-values*) are displayed by user selection.

**Table 3.** Statistical tests used in CNApp.

|  |  | Parametric | Non-parametric |
|---|---|---|---|
| Categoric | $n = 2$ | Student's T-test | Wilcoxon rank-sum test |
|  | $n > 2$ | ANOVA | ANOVA: Kruskal test |
| Numerical |  | Pearson's rank correlation | Spearman's rank correlation |

$n$ = groups defined by annotation variable.

## Clustering analysis

Correlation and clustering methods can be applied into heatmap plots in CNApp *Region profile* section. Accepted correlation methods are Pearson's, Spearman's and Kendall's tests. Hierarchical cluster analysis is applied when clusters are computed in heatmaps.

## Machine learning-based classifier models

We used the *randomForest* R package (RandomForest Package in R, RRID:SCR_015718) (*Liaw and Wiener, 2002*) to compute machine learning classifier models. Variables to define sample groups must be selected, as well as at least one classifier variable. Model construction is performed 50-times and training set is changed by iteration. In order to compute model and select training set, multiple steps and conditions have to be accomplished:

i.   Total $N$ samples divided by $G$ groups depicted by group-defining variable must be higher than n samples from the smaller group:

$$P = \frac{N}{G} \ ; \ P > n$$

ii.  If condition above is not accomplished, then $P$ is set to 75% of n:

$$\text{if } P \leq n \quad \text{then} \quad P = n \cdot 0.75$$

iii. $P$ term must be higher than one, and $N$ must be equal or higher than 20:

$$P > 1 \ \text{ or } \ N \geq 20$$

iv.  Classifier variables, when categorical, shall not have higher number of tags ($Z$) than groups defined ($G$) by group-defining variable:

$$Z < G$$

v.   Training set ($T$) is computed and merged for each group ($g$) from groups ($G$) defined by group variable, extracting $P$ samples from $g$ as follows:

$$t(g) = P \text{ samples from } g \qquad\qquad T = \sum_{i=1}^{g} t_i$$

After model computation, contingency matrix with prediction and reference values by group is created to compute accuracy, specificity and sensitivity by group.

## Acknowledgements

The authors thank Dr. Rodrigo Dienstmann from Vall d'Hebron Institute of Oncology, Barcelona (Spain) for providing CMS and clinical information for the subset of samples included in the COAD cohort from TCGA, and Prof. Ragnhild A Lothe and Dr. Anita Sveen from the Institute for Cancer Research, Oslo University Hospital, Oslo, (Norway) for kindly testing CNApp in an independent validation cohort. We are also thankful to Dr. Laia Codó and Dr. Salvador Capella-Gutierrez from Barcelona Supercomputing Center-Centro Nacional de Supercomputación (BSC-CNS) and the Spanish National Bioinformatics Institute (INB) to allow the usage of the BSC-CNS server as external hosting

for CNApp, and Mr. Ernest Costa and Mr. Rafael Quintero from IDIBAPS/Hospital Clínic for IT support. Part of this work was carried out at the Esther Koplowitz Centre, Barcelona.

## Additional information

### Competing interests

Josep Maria Llovet: is receiving research support from Bayer HealthCare Pharmaceuticals, Eisai Inc, Bristol-Myers Squibb and Ipsen, and consulting fees from Eli Lilly, Bayer HealthCare Pharmaceuticals, Bristol-Myers Squibb, EISAI Inc, Celsion Corporation, Exelixis, Merck, Ipsen, Glycotest, Navigant, Leerink Swann LLC, Midatech Ltd, and Nucleix. The other authors declare that no competing interests exist.

### Funding

| Funder | Grant reference number | Author |
| --- | --- | --- |
| Centro de Investigación Biomédica en Red en el Área temática de Enfermedades Hepáticas y Digestivas | | Sebastià Franch-Expósito<br>Juan José Lozano<br>Antoni Castells<br>Josep Maria Llovet |
| Generalitat de Catalunya | AGAUR 2016BP00161 | Laia Bassaganyas |
| Generalitat de Catalunya | AGAUR 2018FI B1_00213 | Marcos Díaz-Gay |
| Spanish National Health Institute | FPI BES-2017-081286 | Roger Esteban-Fabró |
| European Commission | PCIG11-GA-2012-321937 | Jordi Camps |
| Instituto de Salud Carlos III-European Regional Development Fund | CP13/00160 | Jordi Camps |
| CERCA Program | | Juan José Lozano<br>Antoni Castells<br>Josep Maria Llovet<br>Sergi Castellvi-Bel<br>Jordi Camps |
| Generalitat de Catalunya | 2017 SGR 1035 | Jordi Camps |
| PERIS Generalitat de Catalunya | SLT002/16/00398 | Sergi Castellví-Bel<br>Jordi Camps<br>Antoni Castells |
| Fundación Científica Asociación Española Contra el Cáncer | GCB13131592CAST | Antoni Castells<br>Sergi Castellví-Bel<br>Jordi Camps |
| Horizon 2020 | HEPCAR Ref. 667273-2 | Josep Maria Llovet |
| U.S. Department of Defense | CA150272P3 | Josep Maria Llovet |
| National Cancer Institute | P30-CA196521 | Josep Maria Llovet |
| Samuel Waxman Cancer Research Foundation | | Josep Maria Llovet |
| Spanish National Health Institute | SAF2016-76390 | Josep Maria Llovet |
| Generalitat de Catalunya/AGAUR | SGR-1162 | Josep Maria Llovet |
| Generalitat de Catalunya/AGAUR | SGR-1358 | Josep Maria Llovet |
| Instituto de Salud Carlos III-European Regional Development Fund | PI14/00783 | Jordi Camps |

| | | |
|---|---|---|
| Instituto de Salud Carlos III-European Regional Development Fund | PI17/01304 | Jordi Camps |
| Instituto de Salud Carlos III-European Regional Development Fund | PI17/00878 | Sergi Castellví-Bel |
| Generalitat de Catalunya | 2017 SGR 21 | Antoni Castells |
| Generalitat de Catalunya | 2017 SGR 653 | Sergi Castellví-Bel |

The funders had no role in study design, data collection and interpretation, or the decision to submit the work for publication.

## Author contributions

Sebastià Franch-Expósito, Conceptualization, Resources, Data curation, Software, Formal analysis, Validation, Investigation, Visualization, Methodology; Laia Bassaganyas, Conceptualization, Resources, Data curation, Software, Validation, Investigation, Methodology; Maria Vila-Casadesús, Conceptualization, Software, Methodology; Eva Hernández-Illán, Resources, Validation, Investigation; Roger Esteban-Fabró, Validation; Marcos Díaz-Gay, Conceptualization, Software, Validation, Visualization; Juan José Lozano, Resources, Software, Supervision, Validation, Methodology; Antoni Castells, Supervision, Funding acquisition, Project administration; Josep Maria Llovet, Supervision, Funding acquisition, Investigation, Visualization; Sergi Castellví-Bel, Resources, Supervision, Funding acquisition, Visualization, Project administration; Jordi Camps, Conceptualization, Resources, Supervision, Funding acquisition, Validation, Visualization, Methodology, Project administration

## Author ORCIDs

Sebastià Franch-Expósito ⓘ https://orcid.org/0000-0002-4542-1701
Marcos Díaz-Gay ⓘ http://orcid.org/0000-0003-0658-0467
Antoni Castells ⓘ http://orcid.org/0000-0001-8431-2033
Jordi Camps ⓘ https://orcid.org/0000-0003-2929-4228

## Decision letter and Author response

Decision letter https://doi.org/10.7554/eLife.50267.sa1
Author response https://doi.org/10.7554/eLife.50267.sa2

# Additional files

## Supplementary files

• Supplementary file 1. Comparative analysis of broad CNA frequencies between CNApp and GISTIC2.0. This table displays the comparison between GISTIC and CNApp in terms of broad CNAs by dividing the genome in chromosome arms. First set of columns shows GISTIC frequencies of amplifications (Amp_Freq) and deletions (Del_Freq) with a threshold set at |0.1|. Remaining columns include results from CNApp, using different parameters. CNApp default CNA thresholds are |0.2|, whereas CNApp relaxed CNA thresholds were dropped to |0.1|. CNApp resegmentation parameters (reseg) are indicated accordingly. (XLSX 15 KB)

• Supplementary file 2. Frequencies of focal alterations identified by CNApp using minor sub-cytobands genomic windows. First three columns (No CNA, CN Gain, CN Loss) display the number of samples with no CNA, gains or losses at each minor sub-cytoband using CNApp default thresholds. Next columns ('low-level' amplitude thresholds) indicate frequencies for each alteration (FreqGain, FreqLoss) using CNApp default thresholds. Percentile 90th indicates if this region is among the top recurrent altered regions. The last set of columns ('Moderate' amplitude thresholds) refers to frequencies of gains and losses when amplitude thresholds are set at log2 values of 0.58 (gains) and $-1$ (losses). (XLSX 70 KB)

• Supplementary file 3. Comparison between significant regional peaks of amplification and deletion identified by GISTIC2.0 and CNApp in the LIHC cohort. Statistically significant focal genomic peaks identified by GISTIC2.0 (Q-value and residual Q-value) and the corresponding frequency reported

by CNApp in the equivalent minor sub-cytoband genomic windows. Percentile 90th indicates if this region is among the top recurrent altered regions. (XLSX 16 KB)

- Transparent reporting form

## Data availability

Data and plots presented in the submission were generated by using our CNApp tool. Source code and additional files can be found at GitHub (https://github.com/ait5/CNApp; copy archived at https://github.com/elifesciences-publications/CNApp).

The following previously published datasets were used:

| Author(s) | Year | Dataset title | Dataset URL | Database and Identifier |
|---|---|---|---|---|
| Cancer Genome Atlas Research Network | 2017 | Comprehensive and Integrative Genomic Characterization of Hepatocellular Carcinoma | https://portal.gdc.cancer.gov/projects/TCGA-LIHC | GDC Data Portal, TCGA-LIHC |
| The Cancer Genome Atlas Network | 2012 | Comprehensive Molecular Characterization of Human Colon and Rectal Cancer | https://portal.gdc.cancer.gov/projects/TCGA-COAD | GDC Data Portal, TCGA-COAD |
| The Cancer Genome Atlas Research Network | 2013 | The Cancer Genome Atlas Pan-Cancer analysis project | https://www.cancer.gov/about-nci/organization/ccg/research/structural-genomics/tcga | National Cancer Institute, TCGA-Pancacner |

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
