## [Decision Letter]

**Acceptance summary:**

CNApp is an interactive tool that allows users to visualise whole-genome copy-number profiles from data generated from a number of different platforms, and facilitates performing different analyses such as computation of copy-number alteration scores and sample classification to aid biological interpretation of results. We think it is of utmost importance to provide users with tools for interpreting large amounts of data in an easy manner, and here it is done supplying extensive support for how to use such software. We believe that bench biologists and medical professionals will be interested in this work, as well as bioinformaticians and data analysts.

**Decision letter after peer review:**

Thank you for submitting your article "CNApp: quantification of copy number alterations in cancer and integrative analysis to unravel clinical implications" for consideration by *eLife*. Your article has been reviewed by three peer reviewers, one of whom is a member of our Board of Reviewing Editors, and the evaluation has been overseen by Kathryn Cheah as the Senior Editor. The reviewers have opted to remain anonymous.

The reviewers have discussed the reviews with one another and the Reviewing Editor has drafted this decision to help you prepare a revised submission.

Summary:

In this manuscript, Franch-Expósito et al. describe a new app, CNApp, that facilitates the analysis of large datasets of copy number variation. As a proof of principle, they apply it to TCGA SNP-chip datasets and do three different analyses (overall TCGA, hepatocellular carcinoma and colon cancer). Overall, the reviewers find that the app seems to work, and that the analysis performed is reasonable. In broad terms, they show concordance with other published results, with some differences. However, a number of essential revisions must be performed before the work can be accepted for publication.

Essential revisions:

1) Two of the reviewers mentioned problems when using CNApp in different browsers (one mentioned problems with Safari and Firefox, the other one with Chrome and Safari). Please make sure that the app works with all major browsers (preferably), or at the very least indicate the preferred settings for using it.

2) The major advantage that CNApp would offer is the ability to perform large-scale analyses of copy number variation easily for non-bioinformaticians. However, in the current manuscript it is only compared against GISTIC2.0, but even though there seem to be substantial differences between the results of both CN callers, no effort seems to have been made to indicate that CNApp's calls are more accurate, or why users should trust its output. Please compare the results of CNApp not only against GISTIC2.0 but also against other copy number callers, such as CoNVaQ (Larsen et al., 2018) and CNspector (Markham, Sci Rep, 2019), and offer an analysis of the accuracy of CNApp. The use of simulated datasets is suggested for this purpose (For example, see Mermel et al., 2011).

3) Please justify the boundary between focal and broad CNAs, and why that particular boundary was chosen. On the same note, a BCS value of 4 is suggested to be able to distinguish between tumour subtypes in section "Classification of colon cancer according to CNA scores and genomic regions", however no validation in any external datasets is shown to indicate that this is a reliable value. Can the authors please explain if we should expect to see this replicated in other datasets?

4) Please justify the choice of a random forest classification model, and offer a clear explanation of when this analysis is suitable. The ability to "establish associations between the burden of genomic alterations and any clinical or molecular variable" in a user's dataset could lead to spurious correlations and false positive results, given that the tool is aimed at non-experts.

5) Please explain how the CNApp tool deals with whole-genome duplication events.

[Editors' note: further revisions were suggested prior to acceptance, as described below.]

Thank you for resubmitting your work entitled "CNApp, a tool for the quantification of copy number alterations and integrative analysis revealing clinical implications" for further consideration by *eLife*. Your revised article has been evaluated by Kathryn Cheah (Senior Editor) and a Reviewing Editor.

The manuscript has been improved but there are some remaining issues that need to be addressed before acceptance, as outlined below:

1) In the subsection “, please change the word 'capacity' for 'ability'.

2) Please add labels for the Y-axis in all three plots in Figure 2A.

3) Please be more specific when describing the data shown in supplementary files, adding a description of what columns mean.

4) In legend for Figure 4B, do you mean Wilcoxon rank sum test?

5) In the third paragraph of the subsection “Classification of colon cancer according to CNA scores and genomic regions”, do you mean Figure 4—figure supplement 1E? It is not clear that Figure 4—figure supplement 1F refers to what is specified in the text.

---

## [Author Response]

Essential revisions:1) Two of the reviewers mentioned problems when using CNApp in different browsers (one mentioned problems with Safari and Firefox, the other one with Chrome and Safari). Please make sure that the app works with all major browsers (preferably), or at the very least indicate the preferred settings for using it.

We have addressed the issue with browser incompatibilities and established a new URL to access CNApp. The definitive URL is https://tools.idibaps.org/CNApp/. This URL is redirected to the Barcelona Supercomputing Centre-Centro Nacional de Supercomputación (BSC-CNS) (Barcelona), a world leading institution specialized in high performance computing. BSC-CNS provides the external server hosting CNApp. In addition, we have now ensured to use standard SSL certificates for security reasons.

We have tested the new URL with the most commonly used browsers including Firefox, Chrome and Safari, in Windows, MacOS and Linux/Ubuntu operating systems using more than 500 SNP-array samples at once. CNApp runs smoothly in every browser regardless of the operating system. In case of any issue, contact details are shown at CNApp website and at GitHub.

2) The major advantage that CNApp would offer is the ability to perform large-scale analyses of copy number variation easily for non-bioinformaticians. However, in the current manuscript it is only compared against GISTIC2.0, but even though there seem to be substantial differences between the results of both CN callers, no effort seems to have been made to indicate that CNApp's calls are more accurate, or why users should trust its output. Please compare the results of CNApp not only against GISTIC2.0 but also against other copy number callers, such as CoNVaQ (Larsen et al., 2018) and CNspector (Markham, Sci Rep, 2019), and offer an analysis of the accuracy of CNApp. The use of simulated datasets is suggested for this purpose (For example, see Mermel et al., 2011).

We agree with the reviewers that solely comparing CNApp against GISTIC2.0 falls short for benchmarking the tool. One major achievement in the analysis of copy number alterations has been the development of several programs that are able to infer CNA from different genomic platforms. CNApp aims at taking advantage of the sufficiently exploited field of CNA calling, and performs integrative and comprehensive downstream analysis in order to easily visualize and assess the biological and clinical impact of genomic events. To our knowledge, currently it does not exist any other tool performing such analysis, and thus comparing the complete package of CNApp with other tools individually can only be achieved in an analysis-by-analysis manner. Therefore, similar to what we reported for GISTIC2.0 (i.e., comparison of recurrent CNAs in a cohort), we have now incorporated an additional comparative analysis using CoNVaQ to evaluate the ability of our tool to identify CNAs that are overrepresented in a specific subset of samples. Briefly, below there is a description of the main advantages of using CNApp.

The rationale of identifying recurrent CNAs in a given tumor type relies on the notion that these may have an important (or driver) role in tumorigenesis. GISTIC2.0 represents the gold-standard approach for such analysis and has been largely used as a gene-centered tool to determine potential driver genes, which expression is driven by copy number changes. Similar to GISTIC2.0, using adjustable genomic windows throughout the genome, CNApp also determines the most frequently altered regions in a particular set of samples (described in detail in the subsection “Characterization of cancer types based on CNA scores”), thus uncovering genomic regions with candidate cancer-related genes. As explained in the body of the manuscript, GISTIC2.0 and CNApp results highly overlap, especially when choosing genomic windows in between 1 to 5 Mb and lowering the cutoff. In addition to CNA profiling, CNApp also establishes a comprehensive list of CNAs per sample in order to confidently estimate CNA burdens, for both broad and focal events.

CoNVaQ is specifically (and only) designed to compare two sets of CNV segments (corresponding to two different sample sets) and to assess significant differences associated with one set of samples compared to the other. The main strength of CoNVaQ is that the statistical analysis allows the user to query for specific thresholds and to calculate a q-value by repeated permutation of the samples. This method allows to capture CNVs that using traditional statistical models would not be identified. Likewise, CNApp determines significance based on regression analysis and statistical tests such as Student's t-test or Fisher's exact test in the "Descriptive Regions" section, which allow association of genomic regions based on the seg.mean value or the presence/absence of alterations with specific samples, respectively. Furthermore, CNApp utilizes the Mann-Whitney's test to establish associations between CNA scores and annotated clinical or molecular variables. Visualization of CoNVaQ output (i.e., individual regions differentially represented in one set of samples compared to another) is limited to the UCSC genome browser, but no frequency plots are obtained. In contrast, CNApp is able to extract genome profiles with recurrently minimal affected regions per sample group. To evaluate CNApp ability to identify CNAs overrepresented in a subset of samples, we compared the HCC and COAD TCGA cohorts. To this end, we randomly selected 50 LIHC and 50 COAD-CMS2 samples (CMS subclass with more consistent CNAs), and we compared both sets using CNApp “Descriptive Regions” and CoNVaQ. One important aspect to consider here is that in contrast of CoNVaQ, CNApp does not perform a comparison between each specific CNA but between user-defined genomic windows (arms, half-arms, cytobands, sub-cytobands, and 40, 20, 10, 5 and 1Mb). This is because while CoNVaQ was designed to compare germline copy number changes, which are usually defined by the same breakpoint boundaries among samples, CNApp has been constructed to analyze somatic CNAs, which very rarely show common breakpoints. Thus, we reasoned that the best way to identify differently altered regions between groups of interest would be by using different genomic windows. Taking this into account and in order to perform a comprehensive comparison between CNApp and CoNVaQ, we first used CoNVaQ to determine those regions showing significant differences between COAD and LIHC, defining gained regions when seg.mean ≥0.2 and lost regions when seg.mean ≤-0.2. Fourteen CNAs (5 gains and 9 losses) were identified with a q-value < 0.05. The lengths of these CNAs ranged from 38 to 133 Mb, with an average of 67 Mb, suggesting that they were mainly broad events. Thus, in CNApp we selected the genomic windows comprising chromosome arms, half-arms and cytobands. For consistency, we used the same CNA cut-offs (i.e., |0.2|) and maximum number of base pairs allowed for merging adjacent segments equal to 1,000,000 bp. By using arm-level windows, 9 out of 14 events were detected by CNApp with an adjusted p-value < 0.05 (Fisher’s exact test). For the 5 remaining events, 4 of them were found with a adj p-value between 0.06 – 0.1, and only one CNA (9q) showed p-value >0.1.

This detailed comparison between CNApp and CoNVaQ has been added in the main body of the manuscript (subsection “Identification of recurrent CNAs in hepatocellular carcinoma: Benchmark with other available tools”).

The recently published tool CNspector is intended for visualization of clinically-relevant CNVs derived from NGS workflows, which outperforms other visualization tools such as cnvkit and CopywriteR’s IgV. CNspector usefulness consists in analyzing individual samples and reporting specific CNA possibly associated with a clinical phenotype. Nevertheless, the expected resolution for CNA calls depends exclusively on the segmentation and the NGS enrichment assays. In addition, large-scale studies are not reported for CNspector, thus comparing large sample sets is not intuitive.

Moreover, our tool has been developed in a sequential-like structure, thus performing data analysis using the output generated by CNApp within the same project (e.g., re-segmented data are used to compute CNA scores and region profiles, and subsequently, both CNA scores and genomic region values can be used as proxy variants in the "Classifier model"). While we agree with the reviewers that any new tool should be benchmarked against existing tools performing similar analysis, we emphasize this very unique approach that provides the non-bioinformatic scientific community with the chance to perform comprehensive large-scale genomic analyses. As for the suggestion of using simulated datasets, while we sincerely appreciate the reviewer’s idea, we believe that simulated datasets would be useful when it comes to the point of assessing the performance of tools aiming to specifically infer and call genomic events. Given the fact that CNApp has reproduced very similar results as compared with those from other groups in the genomics filed using TCGA pan-cancer cohorts, it sufficiently proves CNApp reliability.

3) Please justify the boundary between focal and broad CNAs, and why that particular boundary was chosen. On the same note, a BCS value of 4 is suggested to be able to distinguish between tumour subtypes in section "Classification of colon cancer according to CNA scores and genomic regions", however no validation in any external datasets is shown to indicate that this is a reliable value. Can the authors please explain if we should expect to see this replicated in other datasets?

Somatic CNAs can be divided into broad (including whole chromosome and chromosome-arm gains and losses) and focal (genome region-specific amplifications and deletions, smaller than one chromosome arm) events. These two different types of CNAs have different mechanism of origin: while broad events are associated with chromosome mis-segregation during cell division, focal events are mostly generated due to DNA replication and recombination errors. Moreover, recent publications have associated broad and focal CNAs with different molecular and clinical implications. For example, Davoli et al., 2017, identified that cell cycle proliferation markers were present in samples with high burden of focal CNAs, while samples with broad CNAs showed reduced expression of markers for cytotoxic immune cell infiltrates.

Nevertheless, establishing a proper consensus to determine the boundary between broad and focal CNAs is still a challenge (Ben-David U and Amon A 2019 Nat Rev Genetics). For this reason, CNApp offers the user the possibility to customize the boundary and adjust its value to the best fit with the sample set. In the present study, we defined broad events as those affecting more than 50% of the chromosome arm, mimicking the criteria established by Davoli and colleagues (Davoli et al., 2017). By using this threshold, all three scores (BCS, FCS and GCS) showed high correlation (Spearman's rank of 0.975, 0.935 and 0.963, respectively) with values corresponding to the altered fraction of the genome across the TCGA pan-cancer set of 10,635 samples, thus suggesting that the classification between broad and focal events was reasonable. In fact, a systematic comparison using different boundaries might be useful to understand differences in CNA scores and genomic profiles across different sample sets. In our analysis, we tested different boundaries to call for chromosome arm imbalances in the HCC cohort and compared with previously published results. In case the boundary was set at ≥70%, this led to an underestimation of some broad events, while if the boundary was set at ≥40%, the opposite was observed (Figure 3—figure supplement 1).

On the other hand, the analysis of the colon cancer cohort led us to suggest that a BCS value of 4 was be able to distinguish between tumor subtypes, specially between microsatellite deficient (MSI, enriched in CMS1) and proficient (MSS, enriched in CMS2 and 4) tumors. By applying CNApp's "Classifier model" analysis, we chose MSI and MSS status (72 MSI and 225 MSS samples) as defined groups and BCS values as classification proxies to evaluate the prediction results after 50 permutations using the RandomForest R package implemented in CNApp. Training sets of 75% of samples were used in each permutation and the remaining 25% were classified by applying the trained model. Global accuracy in average for the 50-permutation in the RandomForest model was 82.2%. Re-classification of MSI/MSS samples was successfully achieved and distribution of BCS values across MSI and MSS-predicted samples is plotted in Figure 4—figure supplement 1D. Intersection between predicted MSI and MSS groups resulted in a BCS value of 4.75, and further analysis by ROC curve implementation between MSI and MSS groups with BCS values as classifier proxy resulted in an AUC of 0.917 (Author response image 1). In addition, the intersection between sensitivity and specificity from ROC analysis resulted in a BCS value of 3.5 (Figure 4—figure supplement 1E). Altogether, we decided to implement a BCS value of 4 as a threshold to re-classify samples according to their MSI status.

We have included this text in the manuscript to help the reader understanding how the BCS of 4 was achieved and figures as supplementary material (subsection “Classification of colon cancer according to CNA scores and genomic regions”).

**Author response image 1. respfig1:** ROC curve analysis using a BCS value as classifier proxies resulted in AUC = 0.917.

As per the reviewers suggestion, we aimed at validating that a BCS value of 4 was able to predict the MSI status in a completely independent colon cancer cohort. To do so, we contacted the group of Dr. Ragnhild Lothe from Oslo University Hospital in Norway to use the dataset from their recently published article (PMID: 31308487). The validation dataset consisted of segmented data from 147 samples with MSI/MSS status and CMS annotation (MSI, N = 28; MSS, N = 119 and CMS1, N = 27; CMS2, N = 61; CMS3, N = 29; CMS4, N = 30). We then ran CNApp to obtain BCS values for each sample and applied our BCS of 4 as the threshold to re-classify samples (i.e., MSI if BCS ≤4, and MSS if BCS >4). By doing so, we obtained a global accuracy of 81%, which is similar to the accuracy obtained in our initial dataset (82.2%), thus validating that the usage of BCS equal to 4 might be considered as a universal threshold to determine MSI status in colon cancer.

We have added this information in the main body of the manuscript (subsection “Classification of colon cancer according to CNA scores and genomic regions”).

4) Please justify the choice of a random forest classification model, and offer a clear explanation of when this analysis is suitable. The ability to "establish associations between the burden of genomic alterations and any clinical or molecular variable" in a user's dataset could lead to spurious correlations and false positive results, given that the tool is aimed at non-experts.

In the "Classifier Model" section, CNApp implements the randomForest R package to allow researchers assessing the potential use of either genomic features computed by CNApp (e.g., CNA scores) or annotated variables uploaded by the user as classification proxies for clinical and/or molecular data. To do so, CNApp relies on Random Forest, which represents one of the most commonly used supervised machine learning algorithm to build classification and regression models by training decision trees using biomedical annotated data.

Since we are aware that random forest models are not the only algorithms to establish prediction models, we compared the results obtained by CNApp with those after applying the SVM (support vector machine) algorithm. SVM is a supervised learning algorithm highly used in classification tasks as it produces significant accuracies with less computational power. To do so, we randomly generated a training set from the COAD sample set with MSI and MSS annotation, and used it to predict the other 25% of the samples based on the SVM model. This prediction model was generated for 50 permutations and average predictions were chosen, i.e., MSI or MSS status for each sample. Similar to what it has been described before, we selected BCS as a potential classifier between MSI and MSS samples. By doing so, we achieved 84% of global accuracy in our re-classification approach, very close to the accuracy value of 82%, which was reported when implementing Random Forest in CNApp. Furthermore, we were also able to reproduce a density plot for BCS values across MSI and MSS-predicted samples, which further validated a BCS value of 4.7 at the intersection value between MSI and MSS sample groups, as it was previously described using Random Forest (Author response image 2).

**Author response image 2. respfig2:** Density plot of BCS values across MSI and MSS-predicted samples with SVM showing the intersection at 4.7 (blue vertical line).

The fact that both prediction algorithms resulted in very similar performance supports the idea that the classifier model is solid and, at least in this analysis, does not result in spurious correlations. Nevertheless, we understand that the usage of CNApp by non-experts might induce misinterpretations, thus we will reinforce that the output from the "Classifier model" section would need to be further validated using alternative prediction algorithms and/or independent datasets.

5) Please explain how the CNApp tool deals with whole-genome duplication events.

The authors are aware that whole-genome duplication events are rather frequent in cancer. However, whole genome duplication events resulting in a complete set of duplicated chromosomes, which could be masked in the CNA analysis, are rare due to high levels of chromosome mis-segregation events in these samples. Although CNApp does not incorporate the actual ploidy status in the formula to perform re-segmentation, the user might acquire the ploidy through any of the available algorithms (e.g., ASCAT, ABSOLUTE, FACETS or others) and consider it as a variable of analysis. In this way, the user will be able to keep track of the ploidy and at the same time, establish associations with specific genomic features provided by CNApp (e.g., CNA scores, clinical classification) (Author response image 3). A sentence suggesting the possibility to include the ploidy as an annotated variable has been inserted in the revised version of the manuscript (Introduction).

**Author response image 3. respfig3:** Heatmap showing chromosome-arm level CNAs for the COAD cohort with integration of the ploidy values for each sample.

[Editors' note: further revisions were suggested prior to acceptance, as described below.]

The manuscript has been improved but there are some remaining issues that need to be addressed before acceptance, as outlined below:1) In the subsection “, please change the word 'capacity' for 'ability'.

This change has been incorporated in the revised version of the manuscript.

2) Please add labels for the Y-axis in all three plots in Figure 2A.

Labels for Y-axis in the revised Figure 2 have been clarified for all three plots. Moreover, X-axis labels have been also incorporated accordingly.

3) Please be more specific when describing the data shown in supplementary files, adding a description of what columns mean.

Detailed descriptions of the three supplementary files have been added at the end of the main manuscript. Minor changes to the actual files have been incorporated to help the understanding of their content.

4) In legend for Figure 4B, do you mean Wilcoxon rank sum test?

Yes, we meant Wilcoxon rank-sum test. This has been corrected and checked throughout the manuscript.

5) In the third paragraph of the subsection “Classification of colon cancer according to CNA scores and genomic regions”, do you mean Figure 4—figure supplement 1E? It is not clear that Figure 4—figure supplement 1F refers to what is specified in the text.

We apologize for this mistake. Indeed, text is referring to Figure 4—figure supplement 1E in the subsection “Classification of colon cancer according to CNA scores and genomic regions”. This has been corrected accordingly.